# Learning to Infer and Execute 3D Shape Programs

**Yonglong Tian**[†], **Andrew Luo**[†], **Xingyuan Sun**[‡], **Kevin Ellis**[†], **William T. Freeman**[†*],
**Joshua B. Tenenbaum**[†] **& Jiajun Wu**[†]
[†]Massachusetts Institute of Technology
[‡]Princeton University
[*]Google Research
{yonglong,aluo,ellisk,billf,jbt,jiajunwu}@mit.edu
xs5@princeton.edu

## Abstract

Human perception of 3D shapes goes beyond reconstructing them as a set of points or a composition of geometric primitives: we also effortlessly understand higher-level shape structure such as the repetition and reflective symmetry of object parts. In contrast, recent advances in 3D shape sensing focus more on low-level geometry but less on these higher-level relationships. In this paper, we propose *3D shape programs*, integrating bottom-up recognition systems with top-down, symbolic program structure to capture both low-level geometry and high-level structural priors for 3D shapes. Because there are no annotations of shape programs for real shapes, we develop neural modules that not only learn to infer 3D shape programs from raw, unannotated shapes, but also to execute these programs for shape reconstruction. After initial bootstrapping, our end-to-end differentiable model learns 3D shape programs by reconstructing shapes in a self-supervised manner. Experiments demonstrate that our model accurately infers and executes 3D shape programs for highly complex shapes from various categories. It can also be integrated with an image-to-shape module to infer 3D shape programs directly from an RGB image, leading to 3D shape reconstructions that are both more accurate and more physically plausible.

## 1 Introduction

Given the table in Figure 1, humans are able to instantly recognize its parts and regularities: there exist sharp edges, smooth surfaces, a table top that is a perfect circle, and two lower, square layers. Beyond these basic components, we also perceive higher-level, abstract concepts: the shape is bilateral symmetric; the legs are all of equal length and laid out on the opposite positions of a 2D grid. Knowledge like this is crucial for visual recognition and reasoning (Koffka, 2013; Dilks et al., 2011).

Recent AI systems for 3D shape understanding have made impressive progress on shape classification, parsing, reconstruction, and completion (Qi et al., 2017; Tulsiani et al., 2017), many making use of large shape repositories like ShapeNet (Chang et al., 2015). Popular shape representations include voxels (Wu et al., 2015), point clouds (Qi et al., 2017), and meshes (Wang et al., 2018). While each has its own advantages, these methods fall short on capturing the strong shape priors we just described, such as sharp edges and smooth surfaces.

A few recent papers have studied modeling 3D shapes as a collection of primitives (Tulsiani et al., 2017), with simple operations such as addition and subtraction (Sharma et al., 2018). These representations have demonstrated success in explaining complex 3D shapes. In this paper, we go beyond them to capture the high-level regularity within a 3D shape, such as symmetry and repetition.

In this paper, we propose to represent 3D shapes as *shape programs*. We define a domain-specific language (DSL) for shapes, containing both basic shape primitives for parts with their geometric and semantic attributes, as well as statements such as loops to enforce higher-level structural priors.

---

Project page: http://shape2prog.csail.mit.edu

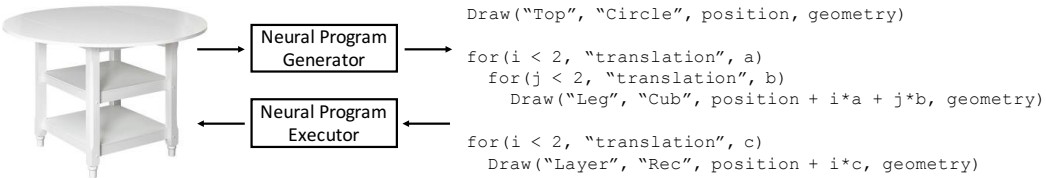

```
Draw("Top", "Circle", position, geometry)

for(i < 2, "translation", a)
  for(j < 2, "translation", b)
    Draw("Leg", "Cub", position + i*a + j*b, geometry)

for(i < 2, "translation", c)
  Draw("Layer", "Rec", position + i*c, geometry)
```

Figure 1: A 3D shape can be represented by a program via a program generator. This program can be executed by a neural program executor to produce the corresponding 3D shape.

Because 3D shape programs are a new shape representation, there exist no annotations of shape programs for 3D shapes. The lack of annotations makes it difficult to train an inference model with full supervision. To overcome this obstacle, we propose to learn a shape program executor that reconstructs a 3D shape from a shape program. After initial bootstrapping, our model can then learn in a self-supervised way, by attempting to explain and reconstruct unlabeled 3D shapes with 3D shape programs. This design minimizes the amount of supervision needed to get our model off the ground.

With the learned neural program executor, our model learns to explain input shapes without ground truth program annotations. Experiments on ShapeNet show that our model infers accurate 3D shape programs for highly complex shapes from various categories. We further extend our model by integrating with an image-to-shape reconstruction module, so it directly infers a 3D shape program from a color image. This leads to 3D shape reconstructions that are both more accurate and more physically plausible.

Our contributions are three-fold. First, we propose 3D shape programs: a new representation for shapes, building on classic findings in cognitive science and computer graphics. Second, we propose to infer 3D shape programs by explaining the input shape, making use of a neural shape program executor. Third, we demonstrate that the inference model, the executor, and the programs they recover all achieve good performance on ShapeNet, learning to explain and reconstruct complex shapes. We further show that an extension of the model can infer shape programs and reconstruct 3D shapes directly from images.

## 2 RELATED WORK

**Inverse procedural graphics.**   The problem of inferring programs from voxels is closely related to inverse procedural graphics, where a procedural graphics program is inferred from an image or declarative specification (Ritchie et al., 2016; Št'ava et al., 2010). Where the systems have been most successful, however, are when they leverage a large shape–component library (Chaudhuri et al., 2011; Schulz et al., 2017) or are applied to a sparse solution space (van den Hengel et al., 2015). Kulkarni et al. (2015a) approached the problem of inverse graphics as inference in a probabilistic program for generating 2D images, or image contours, from an underlying 3D model. They demonstrated results on several different applications using parametric generative models for faces, bodies, and simple multi-part objects based on generalized cylinders. In this work, we extend the idea of inverse procedural graphics to 3-D voxel representations, and show how this idea can apply to large data sets like ShapeNet. We furthermore do not have to match components to a library of possible shapes, instead using a neural network to directly infer shapes and their parameters.

A few recent papers have explored the use of simple geometric primitives to describe shapes (Tulsiani et al., 2017; Zou et al., 2017; Liu et al., 2018), putting the classic idea of generalized cylinders (Roberts, 1963; Binford, 1971) or geons (Biederman, 1987) in the modern context of deep learning. In particular, Sharma et al. (2018) extended these papers and addressed the problem of inferring 3-D CAD programs from perceptual input. We find this work inspiring, but also feel that a key goal of 3-D program inference is to reconstruct a program in terms of semantically meaningful parts and their spatial regularity, which we address here. Some other graphics papers also explore regularity, but without using programs (Mitra et al., 2013; Zhu et al., 2018; Nishida et al., 2018; Li et al., 2017).

Work in the HCI community has also addressed the problem of inferring parametric graphics primitives from perceptual input. For example, Nishida et al. (2016) proposed to learn to instantiate procedural primitives for an interactive modeling system. In our work, we instead learn to instantiate multiple procedural graphics primitives simultaneously, without assistance from a human user.

| Program | $\rightarrow$ | Statement; Program |
|---|---|---|
| Statement | $\rightarrow$ | Draw(Semantics, Shape, Position_Params, Geometry_Params) |
| Statement | $\rightarrow$ | For(For_Params); Program; EndFor |
| Semantics | $\rightarrow$ | semantics 1 \| semantics 2 \| semantics 3 \| ... |
| Shape | $\rightarrow$ | Cuboid \| Cylinder \| Rectangle \| Circle \| Line \| ... |
| Position_Params | $\rightarrow$ | $(x, y, z)$ |
| Geometry_Params | $\rightarrow$ | $(g_1, g_2, g_3, g_4, ...)$ |
| For_Params | $\rightarrow$ | Translation_Params \| Rotation_Params |
| Translation_Params | $\rightarrow$ | (times $i$, orientation $u$) |
| Rotation_Params | $\rightarrow$ | (times $i$, angle $\theta$, axis $a$) |

Table 1: The domain specific language (DSL) for 3D shapes. Semantics depends on the types of objects that are modeled, i.e., semantics for *vehicle* and *furniture* should be different. For details of DSL in our experimental setting, please refer to supplementary.

**Program synthesis.** In the AI literature, Ellis et al. (2018) leveraged symbolic program synthesis techniques to infer 2D graphics programs from images, extending their earlier work by using neural nets for faster inference of low-level cues such as strokes (Ellis et al., 2015). Here, we show how a purely *end–to–end* network can recover 3D graphics programs from voxels, conceptually relevant to RobustFill (Devlin et al., 2017), which presents a purely end-to-end neural program synthesizer for text editing. The very recent SPIRAL system (Ganin et al., 2018) also takes as its goal to learn structured program–like models from (2D) images. An important distinction from our work here is that SPIRAL explains an image in terms of paint-like "brush strokes", whereas we explain 3D voxels in terms of high-level objects and semantically meaningful parts of objects, like legs or tops. Other tangential related work on program synthesis includes Balog et al. (2017); Devlin et al. (2017); Parisotto et al. (2017); Gaunt et al. (2016); Sun et al. (2018a); Liu et al. (2019).

**Learning to execute programs.** Neural Program Interpreters (NPI) have been extensively studied for programs that abstract and execute tasks such as sorting, shape manipulation, and grade-school arithmetic (Reed & De Freitas, 2016; Cai et al., 2017; Bošnjak et al., 2017). In NPI (Reed & De Freitas, 2016), the key insight is that a program execution trace can be decomposed into pre-defined operations that are more primitive; and at each step, an NPI learns to predict what operation to take next depending on the general environment, domain specific state , and previous actions. Cai et al. (2017) improved the generalization of NPIs by adding recursion. Johnson et al. (2017) learned to execute programs for visual question and answering. In this paper, we also learn a 3D shape program executor that renders 3D shapes from programs as a component of our model.

## 3   3D SHAPE PROGRAMS

In this section, we define the domain-specific language for 3D shapes, as well as the problem of shape program synthesis.

Table 1 shows our DSL for 3D shape programs. Each shape program consists of a variable number of program statements. A program statement can be either Draw, which describes a shape primitive as well as its geometric and semantic attributes, or For, which contains a sub-program and parameters specifying how the subprogram should be repeatedly executed. The number of arguments for each program statement varies. We tokenize programs for the purpose of neural network prediction.

Each shape primitive models a semantically-meaningful part of an object. Its geometric attributes (Table 1: Geometry_Params, Position_Params) specify the position and orientation of the part. Its semantic attributes (Table 1: Semantics) specify its relative role within the whole shape (e.g., top, back, leg). They do not affect the geometry of the primitive; instead, they associate geometric parts with their semantic meanings conveying how parts can be shared across object categories semantically and functionally (e.g., a chair and a table may have similar legs).

Our For statement captures high-level regularity across parts. For example, the legs of a table can be symmetric with respect to particular rotation angles. The horizontal bars of a chair may lay out regularly with a fixed vertical gap. Each For statement can contain sub-programs, allowing recursive generation of shape programs.

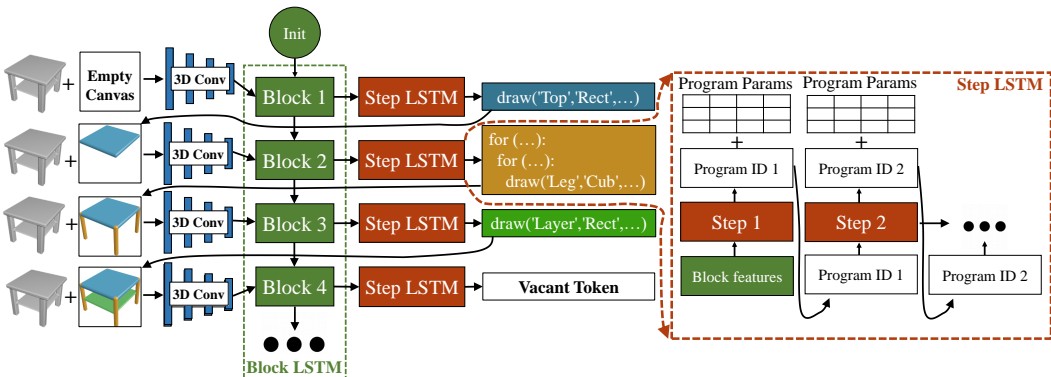

Figure 2: The core of our 3D shape program generator are two LSTMs. The Block LSTM emits features for each program block. The Step LSTM takes these features as input and outputs programs inside each block, which includes either a single drawing statement or compound statements.

The problem of inferring a 3D shape program is defined as follows: predicting a 3D shape program that reconstructs the input shape when the program is executed. In this paper, we use voxelized shapes as input with a resolution of $32 \times 32 \times 32$.

## 4 INFERRING AND EXECUTING 3D SHAPE PROGRAMS

Our model, called *Shape Programs*, consists of a program generator and a neural program executor. The program generator takes a 3D shape as input and outputs a sequence of primitive programs that describe this 3D shape. The neural program executor takes these programs as input and generates the corresponding 3D shapes. This allows our model to learn in a self-supervised way by generating programs from input shapes, executing these programs, and back-propagating the difference between the generated shapes and the raw input.

### 4.1 PROGRAM GENERATOR

We model program generation as a sequential prediction problem. We partition full programs into two types of subprograms, which we call blocks: (1) a single drawing statement describing a semantic part, e.g. circle top; and (2) compound statements, which are a loop structure that interprets a set of translated or rotated parts, e.g. four symmetric legs. This part-based, symmetry-aware decomposition is inspired by human perception (Fleuret et al., 2011).

Our program generator is shown in Figure 2. The core of the program generator consists of two orthogonal LSTMs. The first one,the Block LSTM, connects sequential blocks. The second one, the Step LSTM, generates programs for each block. At each block, we first render the shape described by previous program blocks with a graphics engine. Then, the rendered shape and the raw shape are combined along the channel dimension and fed into a 3D ConvNet. The Block LSTM takes the features output by the 3D ConvNet and outputs features of the current block, which are further fed into the step LSTM to predict the block programs. The reason why we need the step LSTM is that each block might have a different length (*e.g.*, loop bodies of different sizes).

Given block feature $h_{blk}$, the Step LSTM predicts a sequence of program tokens, each consisting of a program id and an argument matrix. The $i$-th row of the argument matrix serves for the $i$-th primitive program. From the LSTM hidden state $h_t$, two decoders generate the output. The softmax classification probability over program sets is obtained by $f_{\text{prog}} : \mathbb{R}^M \to \mathbb{R}^N$. The argument matrix is computed by $f_{\text{param}} : \mathbb{R}^M \to \mathbb{R}^{N \times K}$, where $N$ is the total number of program primitives and $K$ is the maximum possible number of arguments. The feed-forward steps of the Step LSTM are summarized as

$$h_t = f_{\text{lstm}}(x_t, h_{t-1}), \tag{1}$$

$$p_t = f_{\text{prog}}(h_t),\ a_t = f_{\text{param}}(h_t), \tag{2}$$

where the $p_t$ and $a_t$ corresponds to the program probability distribution and argument matrix at time $t$. After getting the program ID, we obtain its arguments by retrieving the corresponding row in the

(a) execute a single drawing statement | (b) execute a compound statement

Figure 3: The learned program executor consits of an LSTM, which encodes multiple steps of programs, and a subsequent 3D DeconvNet which decodes the features to a 3D shape.

argument matrix. At each time step, the input of the Step LSTM $x_t$ is the embedding of the output in the previous step. For the first step, the block feature $h_{blk}$ is used instead.

We pretrain our program generator on a synthetic dataset with a few pre-defined simple program templates. The set of all templates for tables are shown in Section A1. These templates are much simpler than the actual shapes. The generator is trained to predict the program token and regress the corresponding arguments via the following loss $l_{\text{gen}} = \sum_{b,i} w_p l_{\text{cls}}(p_{b,i}, \hat{p}_{b,i})) + w_a l_{\text{reg}}(a_{b,i}, \hat{a}_{b,i})$, where $l_{\text{cls}}(p_{b,i}, \hat{p}_{b,i}))$ and $l_{\text{reg}}(a_{b,i}, \hat{a}_{b,i})$ are the cross-entropy loss of program ID classification and the $\mathcal{L}$-2 loss of argument regression, in step $i$ of block $b$, respectively. The weights $w_p$ and $w_a$ balance the losses between classification and regression.

## 4.2 Neural Program Executor

We propose to learn a neural program executor, an approximate but differentiable graphics engine, which generates a shape from a program. The program executor can then be used for training the program generator by back-propagating gradients. An alternative is to design a graphics engine that explicitly executes a symbolic program to produce a voxelized 3D shape. Certain high-level program commands, such as For statements, will make the executor non-differentiable. Our use of a neural, differentiable executor enables gradient-based fine-tuning of the program synthesizer on unannotated shapes, which allows the model to generalize effectively to novel shapes outside training categories.

Learning to execute a long sequence of programs is difficult, since an executor has to learn to interpret not only single statements but also complex combinations of multiple statements. We decompose the problem by learning an executor that executes programs at the block level, e.g., either a single drawing statement or a compound statements. Afterwards, we integrate these block-level shapes by max-pooling to form the shape corresponding to a long sequence of programs. Our neural program executor includes an LSTM followed by a deconv CNN, as shown in Figure 3. The LSTM aggregates the block-level program into a fixed-length representation. The following deconv CNN takes this representation and generates the desired shape.

To train the program executor, we synthesize large amounts of block-level programs and their corresponding shapes. During training, we minimize the sum of the weighted binary cross-entropy losses over all voxels via

$$\mathcal{L} = \sum_{v \in V} -w_1 y_v \log \hat{y}_v - w_0 (1 - y_v) \log(1 - \hat{y}_v), \tag{1}$$

where $v$ is a single voxel of the whole voxel space $V$, $y_v$ and $\hat{y}_v$ are the ground truth and prediction, respectively, while $w_0$ and $w_1$ balance the losses between vacant and occupied voxels. This training leverages only synthetic data, not annotated shape and program pairs, which is a blessing of our disentangled representation.

## 4.3 Guided Adaptation

A program generator trained only on a synthetic dataset does not generalize well to real-world datasets. With the learned differentiable neural program executor, we can adapt our model to other datasets such as ShapeNet, where program-level supervision is not available. We execute the predicted program by the learned neural program executor and compute the reconstruction loss between the generated shape and the input. Afterwards, the program generator is updated by the gradient back-propagated from the learned program executor, whose weights are frozen.

This adaptation is guided by the learned program executor and therefore called *guided adaptation (GA)*, and is shown in Figure 4. Given an input shape, the program generator first outputs multiple block programs. Each block is interpreted as 3D shapes by the program executor. A max-pooling

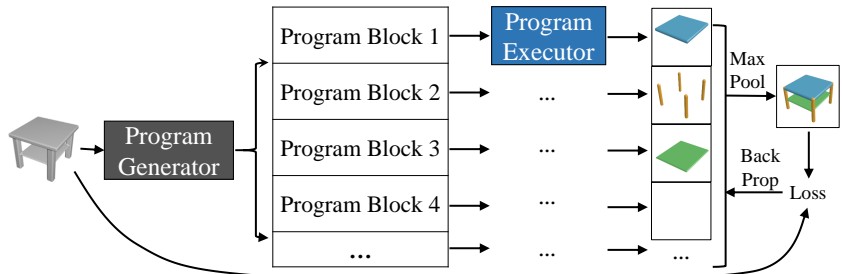

Figure 4: Given an input 3D shape, the neural program executor executes the generated programs. Errors between the rendered shape and the raw input are back-propagated.

operation over these block-level shapes generates the reconstructed shape. The use of max-pooling also enables our executor to handle programs of variable length. Vacant tokens are also executed and pooled. Gradients can then propagate through vacant tokens and the model can learn to add new program primitives accordingly. Here, the loss for *Guided Adaptation* is the summation of the binary cross-entropy loss over all voxels.

## 5 EXPERIMENTS

We present program generation and shape reconstruction results on three datasets: our synthetic dataset, ShapeNet (Chang et al., 2015), and Pix3D (Sun et al., 2018b).

**Setup.** In our experiments, we use a single model to predict programs for multiple categories. Our model is first pretrained on the synthetic dataset and subsequently adapted to target dataset such as ShapeNet and Pix3D under the guidance of the neural program executor. All components of our model are trained with Adam (Kingma & Ba, 2015).

### 5.1 EVALUATION ON THE SYNTHETIC DATASET

**Program generator.** We first pre-train our program generator on our synthetic dataset with simple templates. The synthetic training set includes 100,000 chairs and 100,000 tables. The generator is evaluated on 5,000 chairs and tables. More than 99.9% of the programs are accurately predicted. The shapes rendered by the predicted programs have an average IoU of 0.991 with the input shapes. This high accuracy is due to the simplicity of the synthetic dataset.

**Program executor.** Our program executor is trained on 500,000 pairs of synthetic block programs and corresponding shapes, and tested on 30,000 pairs. The IoU between the shapes rendered by the executor and the ground truth is 0.93 for a single drawing statement and 0.88 for compound statements. This shows the neural program executor is a good approximation of the graphics engine.

### 5.2 GUIDED ADAPTATION ON SHAPENET

**Setup.** We validate the effectiveness of *guided adaptation* by testing our model on unseen examples from ShapeNet. For both tables and chairs, we randomly select 1,000 shapes for evaluation and all the remaining ones for *guided adaptation*.

**Quantitative results.** After our model generates programs from input shapes, we execute these programs with a graphics engine and measure the reconstruction quality. Evaluation metrics include IoU, Chamfer distance (CD) (Barrow et al., 1977), and Earth Mover's distance (EMD) (Rubner et al., 2000). While the pre-trained model achieves 0.99 IoU on the synthetic dataset, the IoU drops below 0.5 on ShapeNet, showing the significant disparity between these two domains. As shown in Table 2, all evaluation metrics suggests improvement after *guided adaptation*. For example, the IoUs of table and chair increase by 0.104 and 0.094, respectively. We compare our method with Tulsiani et al. (2017), which describes shapes with a set of primitives; and CSGNet (Sharma et al., 2018), which learns to describe shapes by applying arithmetic over primitives. For CSGNet, we evaluate two variants: first, CSGNet-original, where we directly test the model released by the original authors;

| Models | IoU ↑ | | CD ↓ | | EMD ↓ | |
|---|---|---|---|---|---|---|
| | table | chair | table | chair | table | chair |
| CSGNet-original | 0.111 | 0.154 | 0.216 | 0.175 | 0.205 | 0.177 |
| Tulsiani et al. (2017) | 0.357 | 0.406 | 0.083 | 0.079 | 0.073 | 0.072 |
| CSGNet-augmented | 0.406 | 0.365 | 0.072 | 0.077 | 0.069 | 0.076 |
| Nearest Neighbour | 0.445 | 0.389 | 0.083 | 0.084 | 0.084 | 0.084 |
| Shape Programs w/o GA | 0.487 | 0.422 | 0.067 | 0.072 | 0.063 | 0.072 |
| Shape Programs | **0.591** | **0.516** | **0.058** | **0.063** | **0.056** | **0.060** |

Table 2: Shape reconstruction results on ShapeNet, evaluated in intersection over union (IoU, higher is better), Chamfer distance (CD, lower is better), and Earth Mover's distance (EMD, lower is better). Our model outperforms the baselines.

second, CSGNet-augmented, where we retrain CSGNet on our dataset with the additional shape primitives we introduced. We also introduce a nearest neighbor baseline, where we use Hamming distance to search for a nearest neighbour from the training set for each testing sample.

Our model without *guided adaptation* outperforms Tulsiani et al. (2017) and CSGNet by a margin, showing the benefit of capturing regularities such as symmetry and translation. The NN baseline suggests that simply memorizing the training set does not generalize well to test shapes. With the learned neural program executor, we try to directly train our program generator on ShapeNet without any pre-training. This trial failed, possibly because of the extremely huge and complicated combinatorial space of programs. However, the initial programs for pre-training can be very simple: e.g., 10 simple table templates (Fig. A1) are sufficient to initialize the model, which later achieves good performance under execution-guided adaptation.

**Qualitative results.** Figure 5 shows some program generation and shape reconstruction results for tables and chairs, respectively. The input shapes can be noisy and contain components that are not covered by templates in our synthetic dataset. After guided adaption, our model is able to extract more meaningful programs and reconstruct the input shape reasonably well.

Our model can be adapted to either add or delete programs, as shown in Figure 5. In (a), we observe an addition of translation describing the armrests. In (b) the "cylinder support" program is removed and a nested translation is added to describe four legs. In (c) and (d), the addition of "Horizontal bar" and "Rectangle layer" leads to more accurate representation. Improvements utilizing modifications to compound programs are not restricted to translations, but can also be observed in rotations, e.g., the times of rotation in (a) is increased from 4 to 5. We also notice new templates emerges after adaptation, e.g., tables in (c) and (d) are not in the synthetic dataset (check the synthetic templates for tables in supplementary). These changes are significant because it indicates the generator can map complex, non-linear relationships to the program space.

## 5.3 STABILITY AND CONNECTIVITY MEASUREMENT

Stability and connectivity are necessary for the functioning of many real-world shapes. This is difficult to capture using purely low-level primitives, but are better suited to our program representations.

We define a shape as stable if its center of mass falls within the convex hull of its ground contacts, and we define a shape as connected if all voxels form one connected component. In Table 3 we compare our model against Tulsiani et al. (2017) and observe significant improvements in the stability of shapes produced by our model when compared to this baseline. This is likely because our model is able to represent multiple identical objects by utilizing translations and rotations. Before GA, our model produces chairs with lower connectivity, but we observe significant improvements with GA. This can be explained by the significant diversity in the ShapeNet dataset under the "chair" class. However, the improvements with GA also demonstrate an ability for our model to generalize. Measured by the percentage of produced shapes that are stable and connected, our model gets significantly better results, and continues to improve with *guided adaptation*.

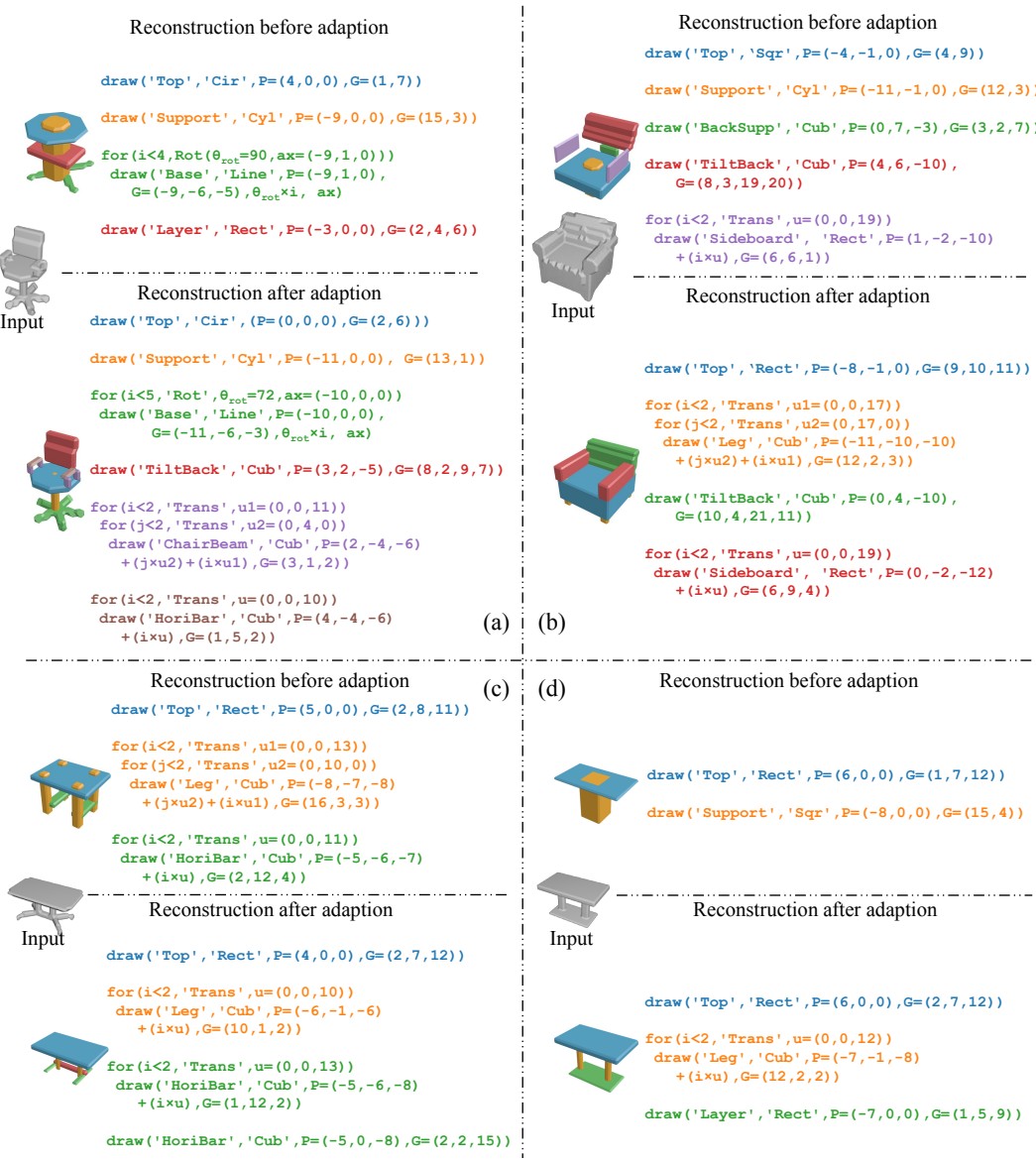

Figure 5: The program generation for ShapeNet chairs and tables. For each shape, the first and second rows represent results before and after *guided adaptation*. Best viewed in color.

| Models | Stable (%) | | Conn. (%) | | Stable & Conn. (%) | |
|---|---|---|---|---|---|---|
| | table | chair | table | chair | table | chair |
| Tulsiani et al. (2017) | 36.7 | 31.3 | 37.1 | **68.9** | 15.4 | 19.6 |
| Shape Programs w/o GA | 94.7 | 95.1 | 76.6 | 54.2 | 73.7 | 51.6 |
| Shape Programs | **97.0** | **96.5** | **78.4** | 68.5 | **77.0** | **66.0** |
| Ground Truth | 98.9 | 97.6 | 98.8 | 97.8 | 97.7 | 95.5 |

Table 3: Measurement of stability and connectivity. Our model is able to capture shape regularity such as symmetry. Therefore, shapes represented by our programs are more stable and better connected.

## 5.4 GENERALIZATION ON OTHER SHAPES

While our program generator is pre-trained only on synthetic chairs and tables, generalization on other shape categories is desirable. We further demonstrate that with *guided adaptation*, our program generator can be transferred to other unseen categories.

| Models | IoU ↑ | | | | CD ↓ | | | |
|---|---|---|---|---|---|---|---|---|
| | bed | sofa | cabinet | bench | bed | sofa | cabinet | bench |
| Shape Programs w/o GA | 0.234 | 0.296 | 0.251 | 0.176 | 0.126 | 0.103 | 0.104 | 0.098 |
| Shape Programs | **0.367** | **0.597** | **0.478** | **0.418** | **0.096** | **0.067** | **0.092** | **0.059** |

Table 4: Shape reconstruction results on unseen categories. Results with or without *guided adaptation* in intersection over union (IoU, higher is better) and Chamfer distance (CD, lower is better).

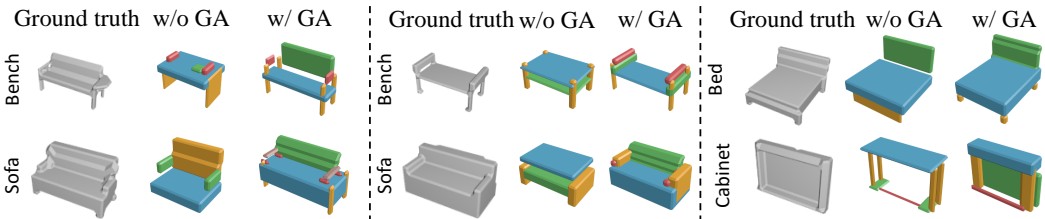

Figure 6: ShapeNet objects from unseen categories reconstructed with shape programs before and after *guided adaptation*. Shape Programs can learn to adapt and explain objects from novel classes.

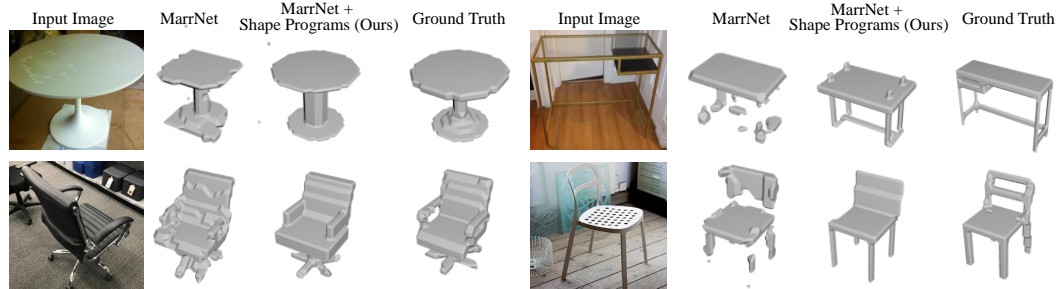

Figure 7: 3D reconstruction results on Pix3D dataset. MarrNet generates fragmentary shapes and our model further smooths and completes such shapes.

We consider *Bed*, *Bench*, *Cabinet*, and *Sofa*, which share similar semantics with table and chair but are unseen during pre-training. We split 80% shapes of each category for *guided adaptation* and the remaining for evaluation. Table 4 suggests the pre-trained model performs poorly for these unseen shapes but its performance improves with this unsupervised *guided adaptation*. The IoU of bed improves from 0.23 to 0.37, sofa from 0.30 to 0.60, cabinet from 0.25 to 0.48, and bench from 0.18 to 0.42. This clearly illustrates the generalization ability of our framework. Visualized examples are show in Figure 6.

### 5.5 SHAPE COMPLETION AND SMOOTHING BY PROGRAMS

One natural application of our model is to complete and smooth fragmentary shapes reconstructed from 2D images. We separately train a MarrNet (Wu et al., 2017) model for chairs and tables on ShapeNet, and then reconstruct 3D shapes from 2D images on the Pix3D dataset. As shown in Figure 7, MarrNet can generate fragmentary shapes, which are then fed into our model to generate programs. These programs are executed by the graphics engine to produce a smooth and complete shape. For instance, our model can complete the legs of chairs and tables, as shown in Figure 7.

While stacking our model on top of MarrNet does not change the IoU of 3D reconstruction, our model produces more visually appealing and human-perceptible results. A user study on AMT shows that 78.9% of the participant responses prefer our results rather than MarrNet's.

## 6 DISCUSSION

We have introduced 3D shape programs as a new shape representation. We have also proposed a model for inferring shape programs, which combines a neural program synthesizer and a neural

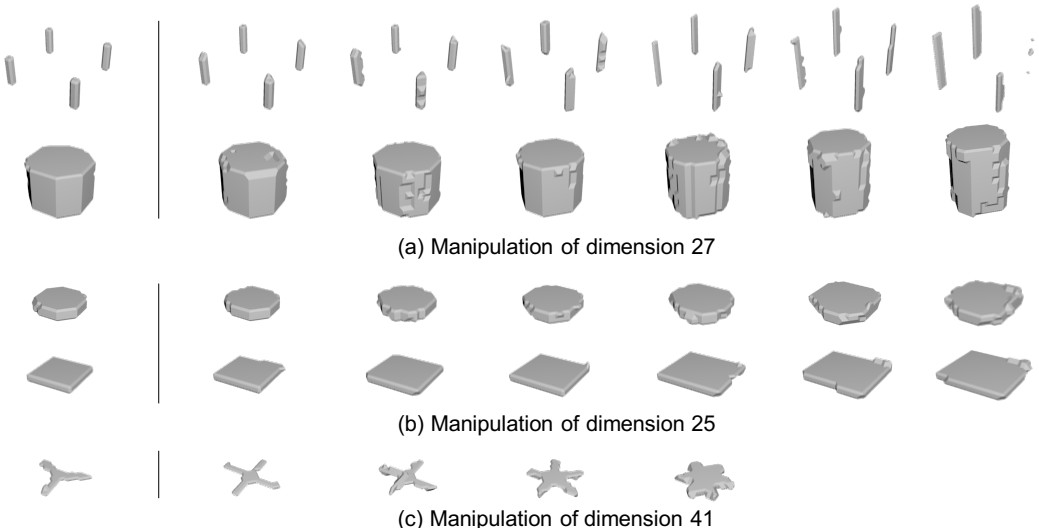

(a) Manipulation of dimension 27

(b) Manipulation of dimension 25

(c) Manipulation of dimension 41

Figure 8: We visualize the effect of manipulating individual dimensions in the intermediate representation of neural program executor. For example, dimension 27 corresponds to the height of primitives, dimension 25 to the radius of primitives, and dimension 41 to the times of primitive repetition.

executor. Experiments on ShapeNet show that our model successfully explains shapes as programs and generalizes to shapes outside training categories. Further experiments on Pix3D show our model can be extended to infer shape programs and reconstruct 3D shapes directly from color images. We now discuss key design choices and future work.

**Analyzing the neural program executor.** We look deep into the intermediate representation of the neural program executor, which is a 64-dimensional vector output by the LSTM (see Figure 3). We manipulate individual dimensions and visualize the generated voxels. Figure 8 shows that these dimensions capture interpretable geometric features (e.g., height, radius, and number of repetitions).

**Design of the DSL.** Our design of the DSL for shape programs makes certain semantic commitments. A DSL with these semantics has advantages and disadvantages: it naturally supports semantic correspondences across shapes and enables better in-class reconstructions; on the other hand, it may limit the ability to generalize to shapes outside training classes. Our current instantiation focuses on the semantics of furniture (a superclass, whose subclasses share similar semantics). Within this superclass, our model generalizes well: trained on chairs and tables, it generalizes to new furniture categories such as beds. In future work, we are interested in learning a library of shape primitives directly from data, which will allow our approach to adapt automatically to new superclasses or domains of shape.

**Structure search *vs*. amortized inference.** For our program synthesis task, we use neural nets for amortized inference rather than structure search, due to the large search space and our desire to return a shape interpretation nearly instantaneously, effectively trading neural net training time for fast inference at test time. Our model takes 5 ms to infer a shape program with a Titan X GPU. We also considered various possible approaches for structured search over the space of shape programs, but decided that these would most likely be too our slow for our goals.

One approach to structured search is constraint solving. Ellis et al. (2015) used the performant Z3 SMT solver (De Moura & Bjørner, 2008) to infer 2D graphics programs, taking 5-20 minutes for problems arguably simpler than our 3D shape programs. Other approaches could be based on stochastic search, such as MCMC in the space of programs. For the related problem of inverse graphics from 2D images, MCMC, like constraint solving, takes too long for perception at a glance (Kulkarni et al., 2015b). Efficient integration of discrete search and amortized inference, however, is a promising future research direction.

**Acknowledgments.** We thank Keyulu Xu and Xiuming Zhang for insightful discussions and anonymous reviewers for their feedback. This work is supported by the Center for Brains, Minds and Machines (NSF #1231216), NSF #1447476, ONR MURI N00014-16-1-2007, and Facebook.

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

## A.1 DEFINED PROGRAMS

The details of semantics and shape primitives in our experimental setting for furniture are shown in Table 5. Due to the semantic nature of objects, while a few semantics are category specific, e.g., "ChairBeam", other semantics are shared across different shape categories, e.g., "Leg" and "Top".

| | | |
|---|---|---|
| | Leg | Chair leg, table leg, etc. Usually long and used jointly for support |
| | Top | Seat top, table top, etc. Usually a broad and flat surface |
| | Layer | Shelf embedded in table, cabinet shelf etc. Usually a flat surface between other similar shapes |
| | Support | Chair support, table support etc. A monolithic object used to raise things off the ground |
| | Base | Base of an sofa, table etc. Usually a flat surface on the ground to help with stability |
| | Sideboard | Sideboard of a cabinet, table etc. A vertical, flat surface on the bottom half of an object |
| Semantics | Horizontal Bar | Horizontal bar of a chair etc. A thin bar used for structural integrity |
| | Vertical Board | Vertical board of a arm rest etc. A vertical, flat surface used by humans for arm support |
| | Locker | Table drawer etc. A boxy object used to put things in |
| | Back | Chair back, Sofa back etc. A surface used for resting backs on |
| | Back support | Office chair back support beam etc. A beam used to support a back rest |
| | ChairBeam | Arm rest support beam in chairs, benches etc. A long object used to support an arm rest |
| | Cylinder (Cyl) | $P = (x, y, z), G = (t, r)$, draw a cylinder at $(x, y, z)$ with sizes $(t, r)$ |
| | Cuboid (Cub) | $P = (x, y, z), G = (t, r_1, r_2, [ang])$, draw a cuboid at $(x, y, z)$ with sizes $(t, r_1, r_2)$ and optional $ang$ of tilt along front/back |
| | Circle (Cir) | $P = (x, y, z), G = (t, r)$, draw a circle at $(x, y, z)$ with sizes $(t, r)$, t is usually small |
| Shapes | Square (Sqr) | $P = (x, y, z), G = (t, r)$, draw a square at $(x, y, z)$ with sizes $(t, r)$, t is usually small |
| | Rectangle (Rect) | $P = (x, y, z), G = (t, r_1, r_2)$, draw a rectangle at $(x, y, z)$ with sizes $(t, r_1, r_2)$, t is usually small |
| | Line (Line) | $P = (x_1, y_1, z_1), G = (x_2, y_2, z_2)$, draw a line from $P$ to $G$ |

Table 5: The list of semantics and shapes, as well as associated parameters used by our model

## A.2 ARCHITECTURE DETAILS

**Program Generator.** The program executor contains a 3D ConvNet and two LSTMS. *(1) 3D ConvNet.* This 3D ConvNet is the first part of the program generator model. It consists of 8 3D convolutional layers. The kernel size of each layer is 3 except for the first one whose kernel size is 5. The number of output channels are (8,16,16,32,32,64,64,64), respectively. The output of the last layer is averaged over the spatial resolution, which gives a 64-dimension embedding. *(2) Block LSTM and Step LSTM.* These two LSTMs share similar structure. Both are one-layer LSTMs. The dimensions of the input and hidden state are both 64.

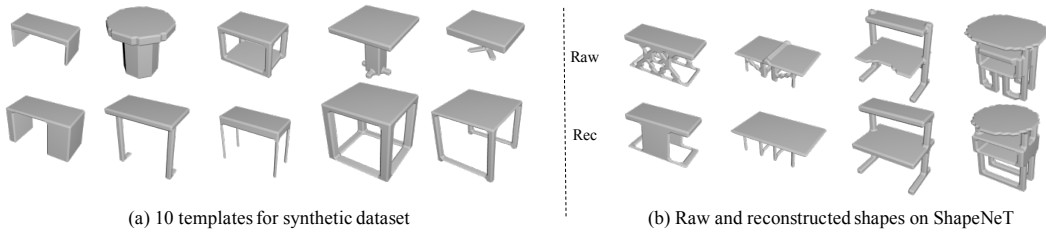

(a) 10 templates for synthetic dataset     (b) Raw and reconstructed shapes on ShapeNeT

Figure A1: (a) shows random samples from all of our 10 table templates for synthetic dataset (b) shows raw and reconstructed tables on ShapeNet

**Program Executor.** The program executor contains an LSTM and a 3D DeConvNet. *(1) LSTM.* This LSTM aggregates a block-level programs into a 64-dimensional vector. The dimension of the hidden state is also 64. The input of each time step is the concatenation of category distribution over programs and the corresponding parameters retrieved from the parameter matrix. *(2) 3D DeConvNet.* It consists of 7 layers. TransposedConv layer with kernel size 4 and Conv layer with kernel size 3 are alternating. The number of output channels are (64, 64, 16, 16, 4, 4, 2), respectively. The output of the last layer is fed into a sigmoid function to generate the 3D voxel shape.

**End-to-end differentiability.** The end-to-end differentiability is obtained via our design of the neural program executor. The output of the program inference model is actually continuous. A real execution engine (not the neural executor) actually contains two steps: (1) discretize such output, and (2) execute the discretized program to generate the voxel. Our neural executor is learned to jointly approximate both steps, thus the whole pipeline can be differentiable in an end-to-end manner.

## A.3 Synthetic Templates v.s. ShapeNet

ShapeNet was proposed to be the ImageNet of shapes; it is therefore highly diverse and intrinsically challenging. Our synthetic dataset were designed to provide minimal, simple guidance to the network. In Figure A1, (a) shows sampled shapes from all of our 10 table templates, while (b) shows the ground truth and reconstructed tables in ShapeNet, which are siginificantly more complex. Such disparity of complexity explains why we saw a dramatic drop of IoU when we directly tested on ShapeNet with model only pretrained on synthetic dataset. Our guided adaptation further adapt the pre-trained model.

## A.4 Additional Results

In Figure A2 through Figure A8, we show the generated shape and programs using a network that is only pretrained jointly on synthetic "table" and "chair" objects, and a network that is pretrained then further enhanced by guided adaptation on ShapeNet data. Figure A2 and Figure A3 correspond to "chairs", Figure A4 to "tables", Figure A5 to "benches", Figure A6 to "couches", Figure A7 to "cabinets", and Figure A8 to "beds". In Figure A2, Figure A3, and Figure A4, even though "chair" and "table" have been seen in the synthetic dataset, we still note improvements in program inference and shape reconstruction on ShapeNet after guided adaptation. This is because our synthetic data is simpler than ShapeNet data. When directly using a model pretrained on synthetic "chairs" and "tables" to other classes, it is not surprised some shapes are interpreted as tables or chairs. However, such mispredictions dramatically drop after our *guided adaptation*.

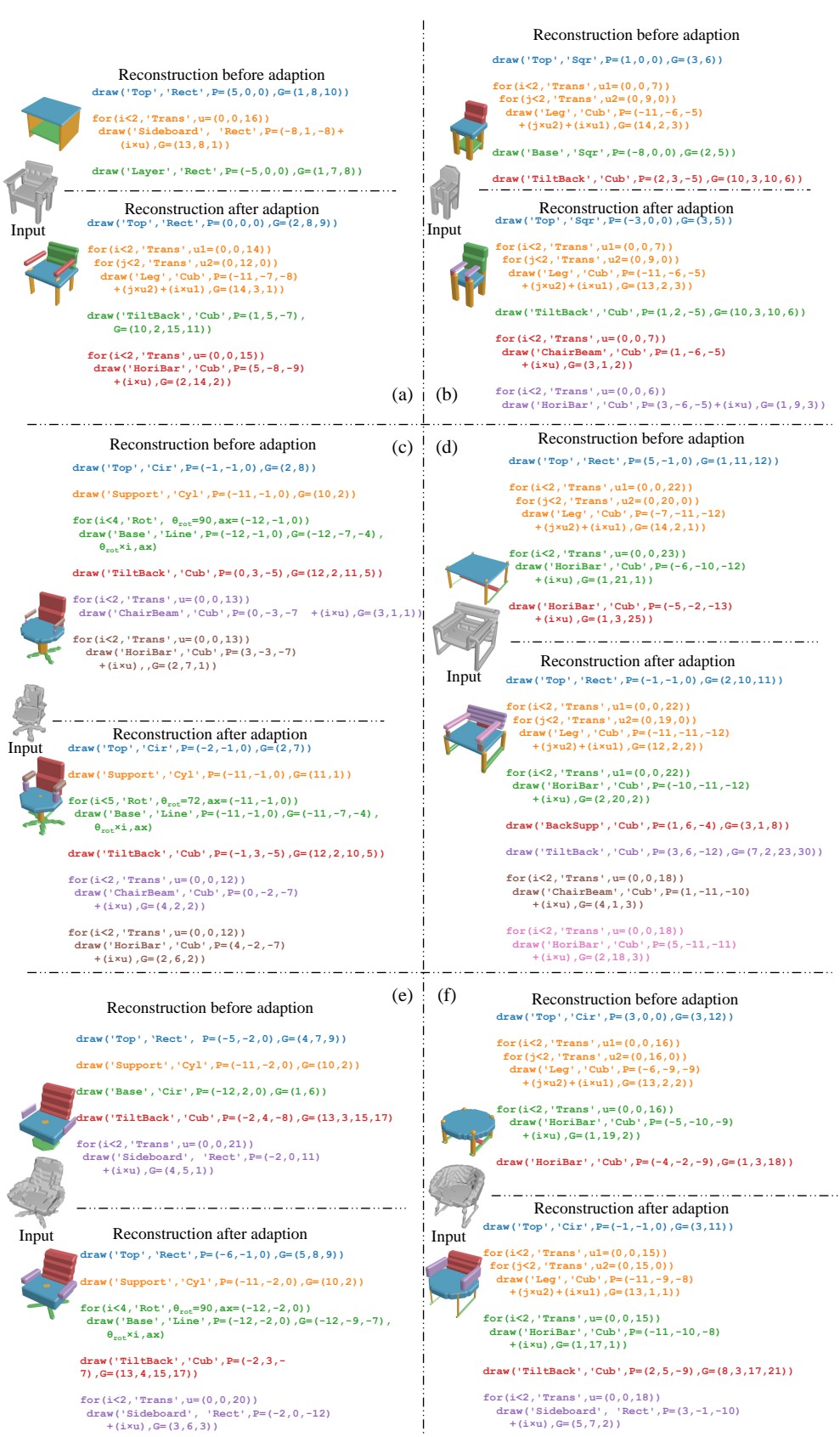

Figure A2: (a) to (f) show the generated shapes and programs for ShapeNet chairs

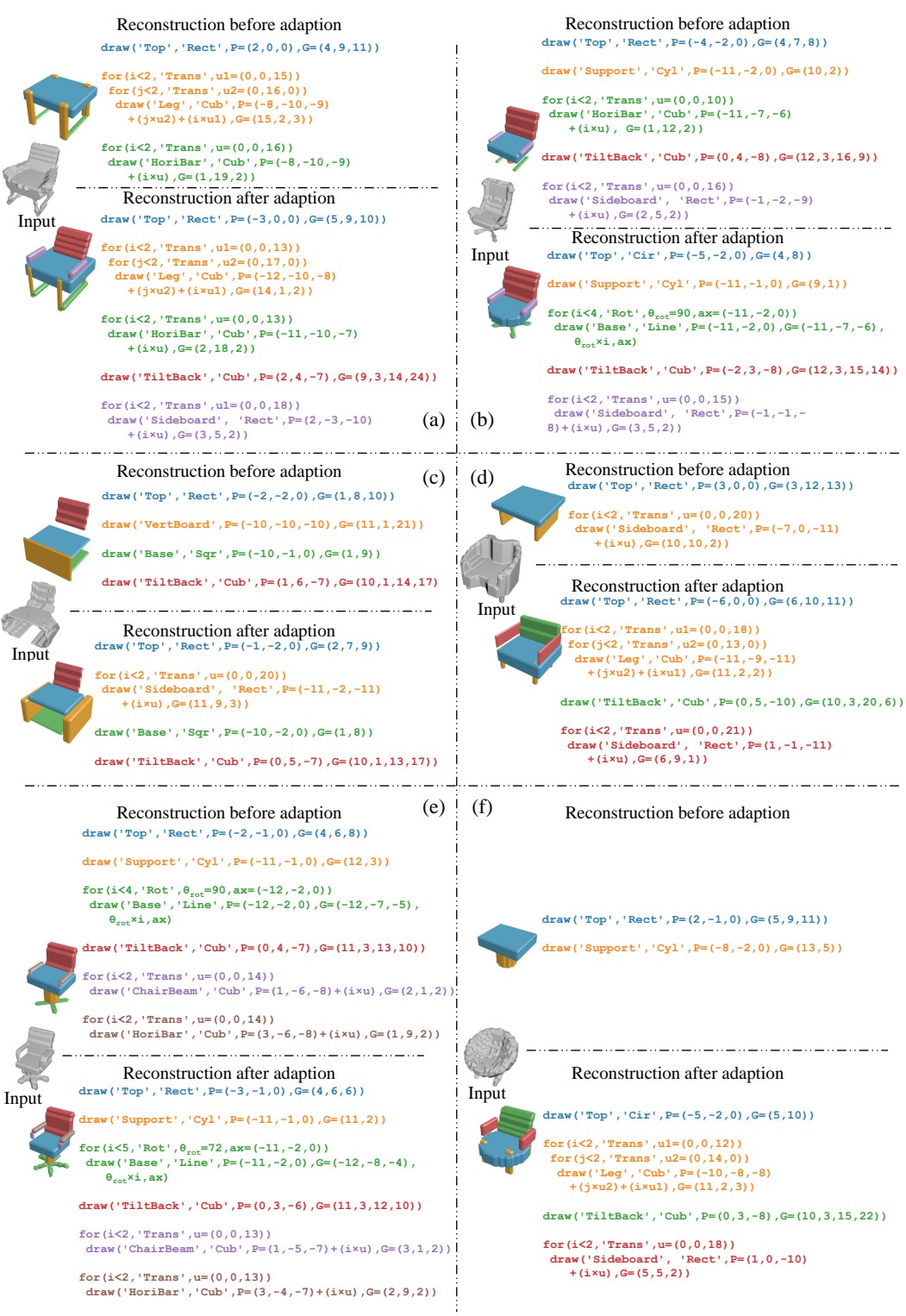

Figure A3: (a) to (f) show the generated shapes and programs for ShapeNet chairs

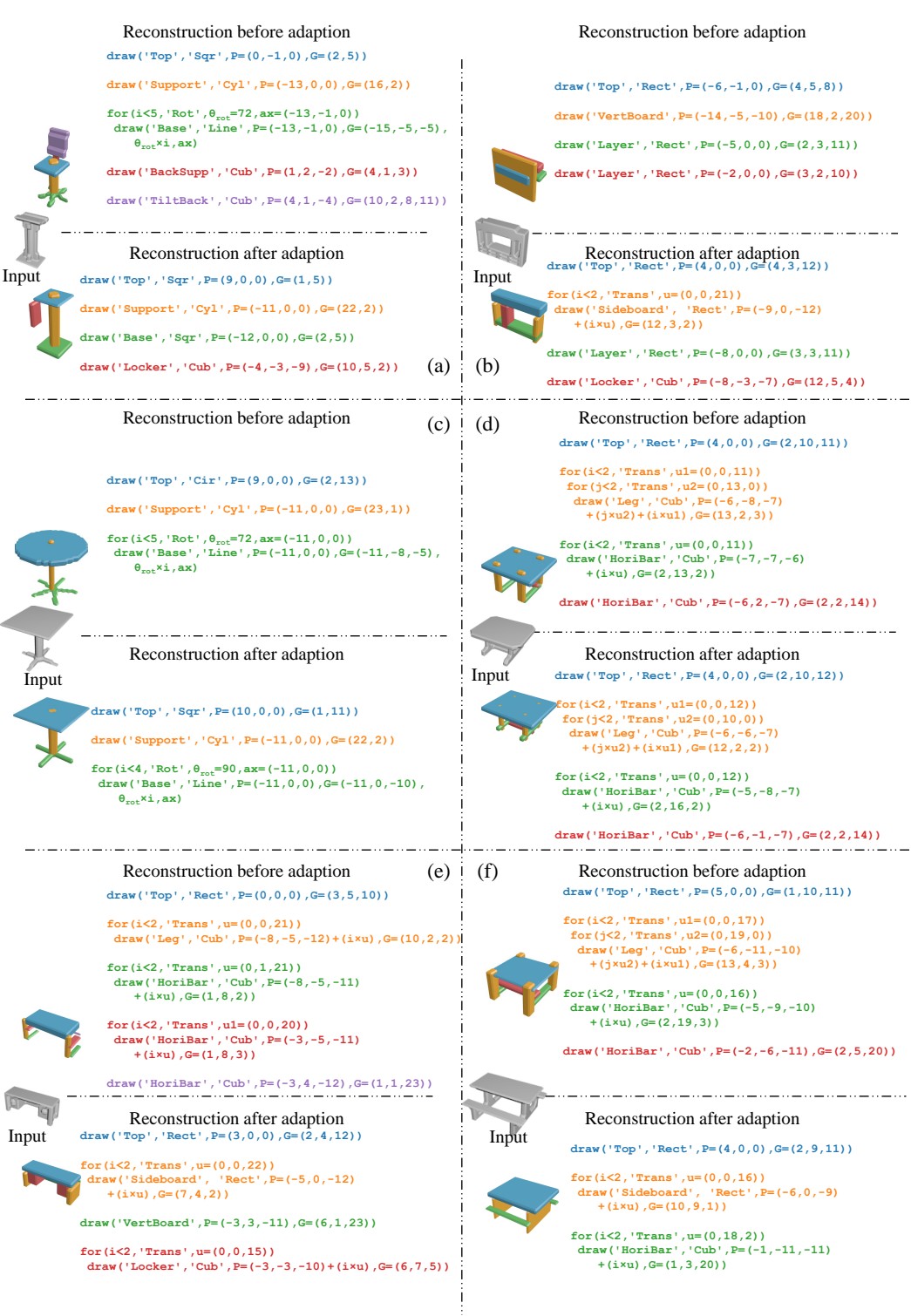

Figure A4: (a) to (f) show the generated shapes and programs for ShapeNet tables

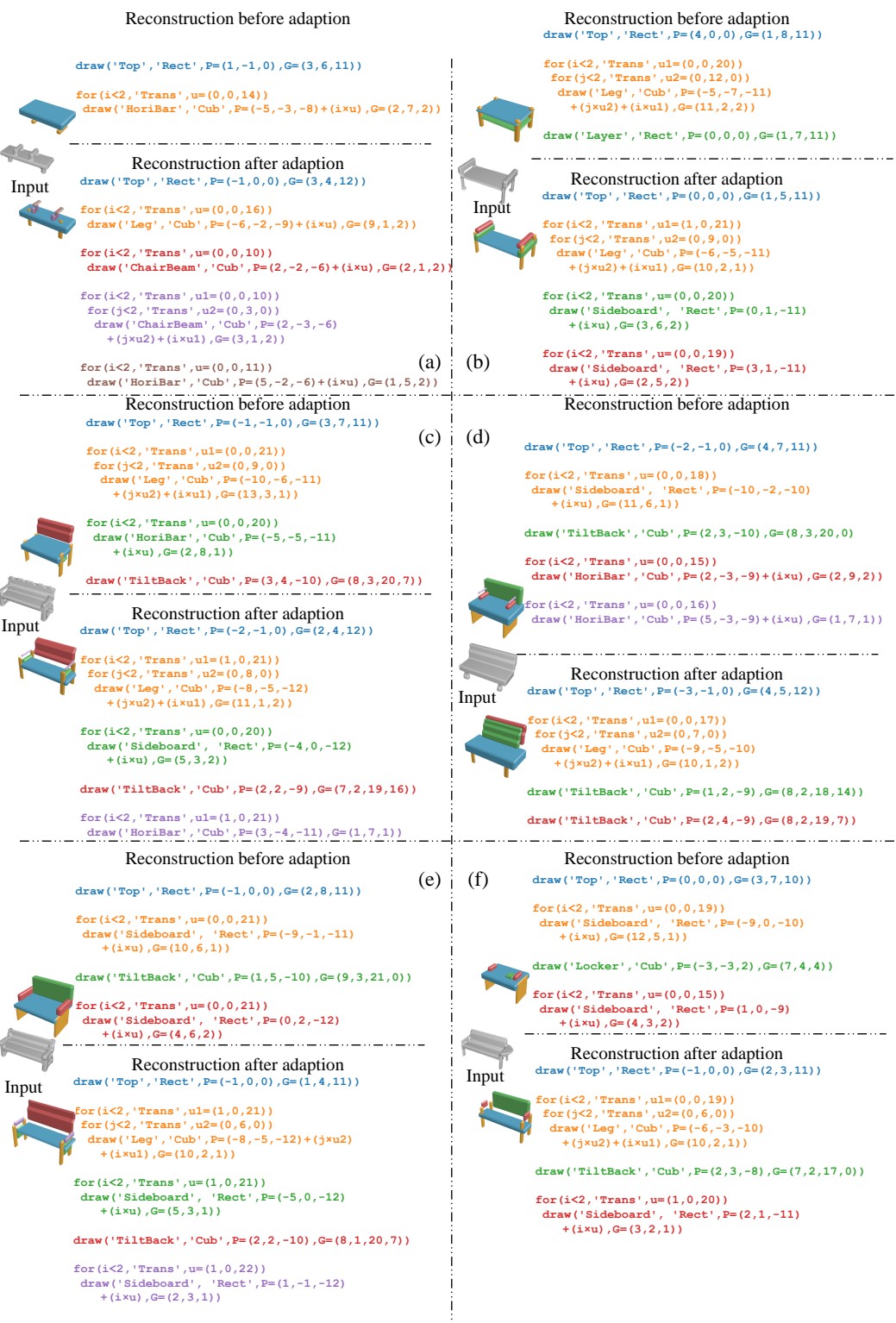

Figure A5: (a) to (f) show the generated shapes and programs for ShapeNet benches

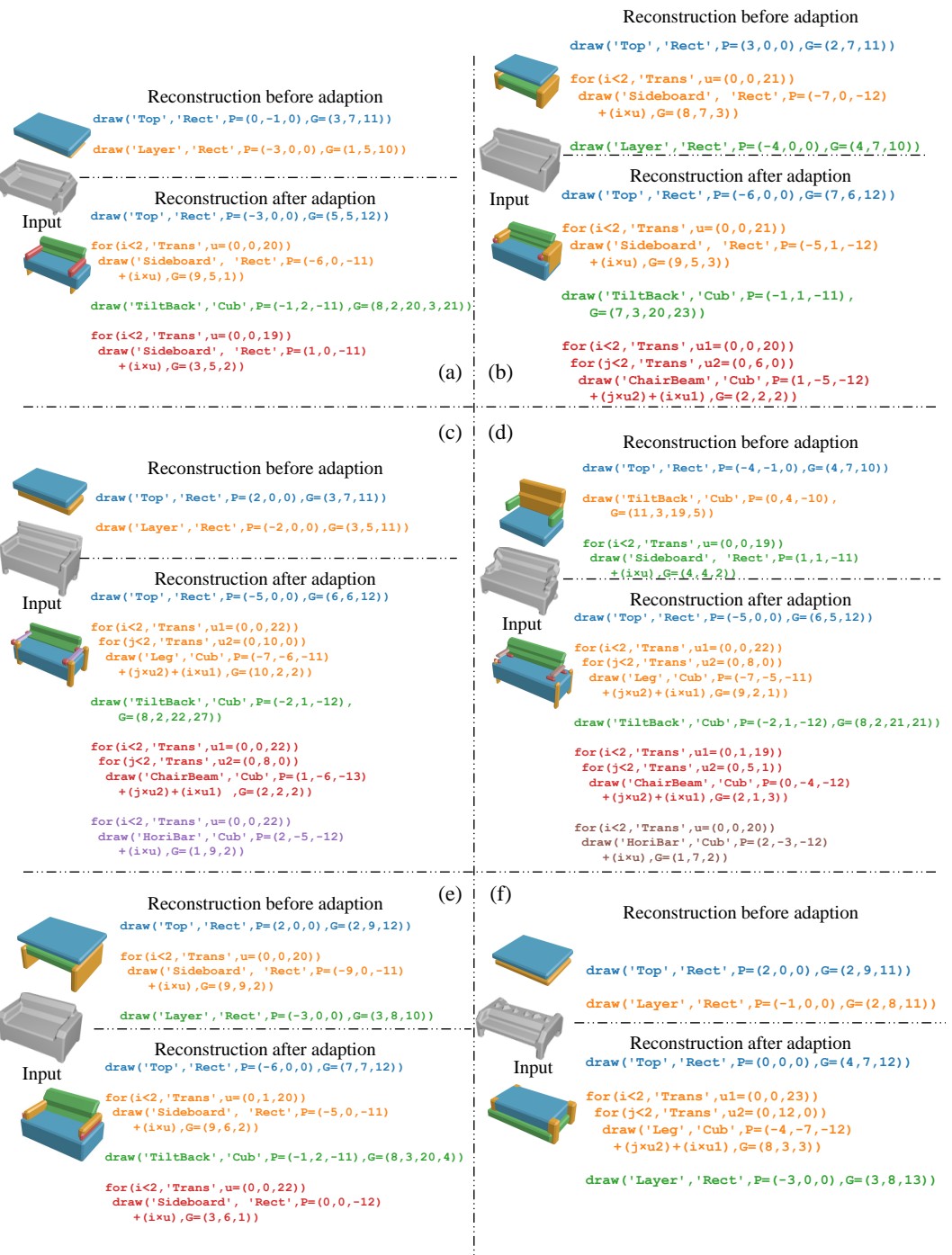

Figure A6: (a) to (f) show the generated shapes and programs for ShapeNet couches

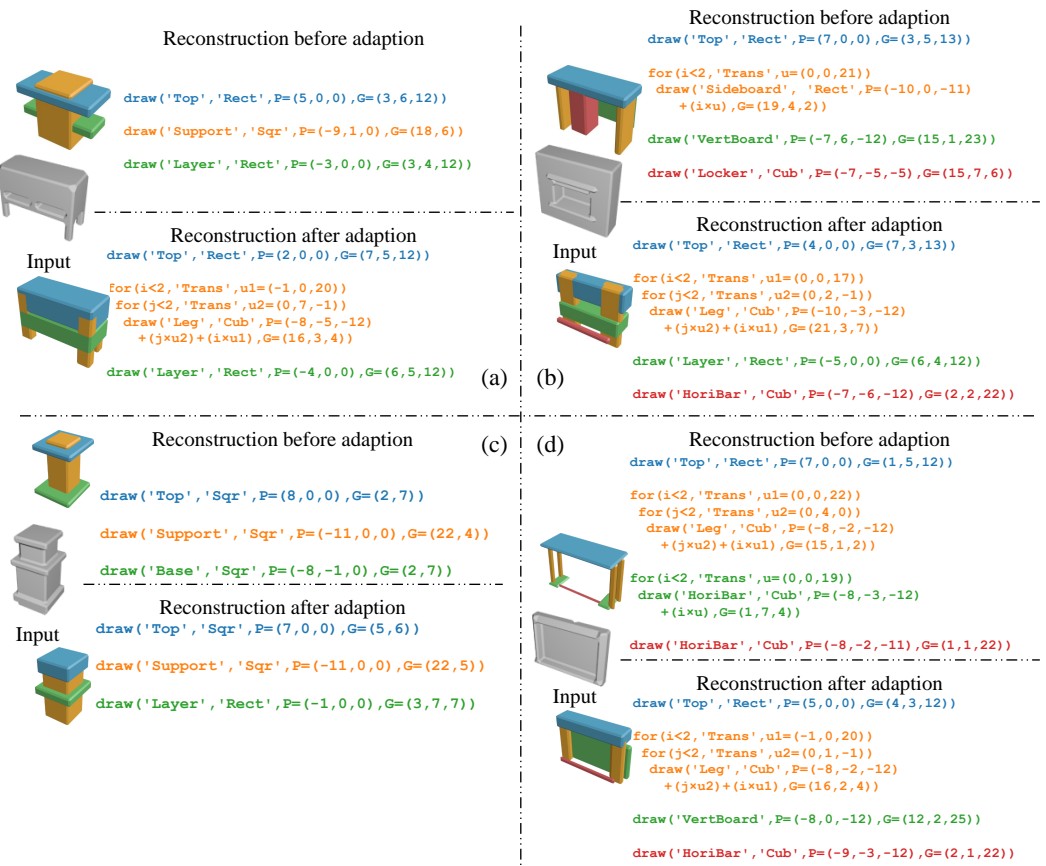

Figure A7: (a) to (d) show the generated shapes and programs for ShapeNet cabinets

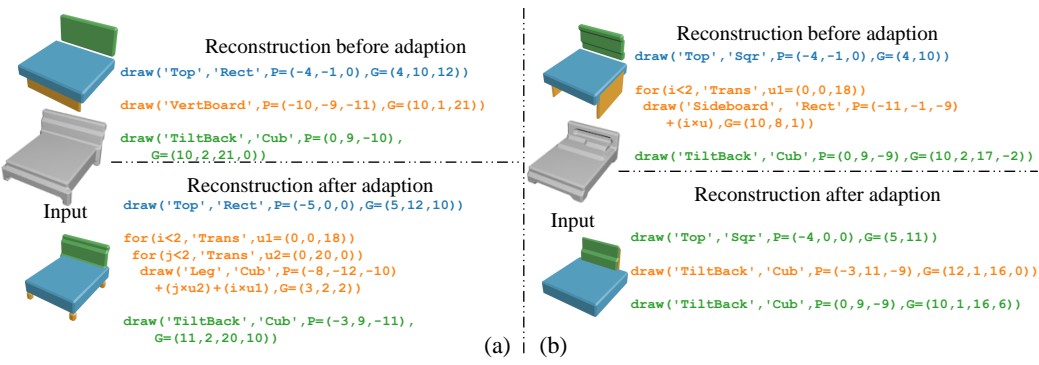

Figure A8: (a) and (b) show the generated shapes and programs for ShapeNet beds

