# OpenReview forum: "Learning to Infer and Execute 3D Shape Programs"
_ICLR.cc/2019/Conference_

### Official Review · AnonReviewer2 · 2018-11-03
**Elegant synthesis approach to a new interesting domain of representing 3D shapes**

**Rating:** 7
**Confidence:** 4

**Review:**

This paper presents a methodology to infer shape programs that can describe 3D objects. The key intuition of the shape programs is to integrate bottom-up low-level feature recognition with symbolic high-level program structure, which allows the shape programs to capture both high-level structure and the low-level geometry of the shapes. The paper proposes a domain-specific language for 3D shapes that consists of “For” loops for capturing high-level regularity, and associates objects with both their geometric and semantic attributes. It then proposes an end-to-end differentiable architecture to learn such 3D programs from shapes using an interesting self-supervised mechanism. The neural program generator proposes a program in the DSL that is executed by a neural program execution module to render the corresponding output shape, which is then compared with the original shape and the difference loss is back-propagated to improve the program distribution. The technique is evaluated on both synthetic and ShapeNet tasks, and leads to significant improvements compared to Tulsiani et al. that embed a prior structure on learning shape representations as a composition of primitive abstractions. In addition, the technique is also paired with MarrNet to allow for a better 3D reconstruction from 2D images.

Overall, this paper presents an elegant idea to describe 3D shapes as a DSL program that captures both geometric and spatial abstractions, and at the same time captures regularities using loops. CSGNet [Sharma et al. 2018] also uses programs to describe 2D and 3D shapes, but the DSL used here is richer as it captures more high-level regularities using loops and also semantic relationships such as top, support etc. The idea of training a neural program executor and using it for self-supervised training is quite elegant. I also liked the idea of guided adaption to make the program generator generalize beyond the synthetic template programs. Finally, the results show impressive improvements and generalization capability of the model.

Can the authors comment on some notion of completeness of the proposed DSL? In other words, is this the only set of operators, shapes, and semantics needed to represent all of ShapeNet objects? Also, it might be interesting to comment more on how this particular DSL was derived. Some of the semantics operator such as “Support”, “Locker”, etc. look overly specific to chair and tables. Is there a way to possibly learn such abstractions automatically?

What is the total search space of programs in this DSL? How would a naive random search perform in this synthesis task?

I also particularly liked the decomposition of programs into draw and compound statements, and the corresponding program generator decomposition into 2 steps BlockLSTM and StepLSTM. At inference time, does the model use some form of beam search to sample block programs or are the results corresponding to top-1 prediction?

Would it be possible to compare the results to the technique presented in CSGNet [Sharma et al. 2018]?  There are some key differences in terms of using lower-level DSL primitives and using REINFORCE for training the program generator, but it would be good to measure how well having higher-level primitives improve the results.

I presume the neural program executor module was trained using a manually-written shape program interpreter. How difficult is it to write such an interpreter? Also, how easy/difficult is to extend the DSL with new semantics operator and then write the corresponding interpreter extension?

Minor typos:
page 3: consists a variable → consists of a variable
page 5: We executes → We execute
page 6: synthetica dataset → synthetic dataset

---

> ### Author Response · Authors · 2018-11-15
> **Response to Reviewer 2**
>
> Thank you for the very constructive comments.
>
> 1. DSL
> The current DSL is designed to represent furnitures. Representing all ShapeNet objects needs a richer set of primitives, e.g., curved cylinders for mug handles. When we design such DSL, the main challenge is on semantics. For humans, some semantics are shared across different object categories, e.g., “top” can be shared by tables and bed, while some are just category-specific, “armrest” is mainly for chairs. Following this spirit, we include both category-specific and shared semantics for the instantialization of furnitures. Learning a primitive library from data is a natural research direction, and we are working on it as follow-up.
>
> 2. Baselines
> We agree that it’s important to add more baselines. In the revision, we will include comparisons with the following three algorithms:
> 1) Nearest neighbors. For a given test shape, we search its nearest neighbor in the training set.
> 2) CSGNet-original (the original model released by the authors of CSGNet)
> 3) CSGNet-augmented (the augmented CSGNet model trained on our dataset with additional shape primitives we introduced).
>
> Amortized inference is essential for our task due to its large search space. Our model takes 5 ms to infer a shape program with a Titan X GPU. There are two possible approaches for a structured search over the space of programs, both of which will be too slow for our task:
> 1) Constraint solving: we would have to use an SMT solver. Ellis et al [1] used SMT solvers to infer 2D graphics programs, and takes on the order of 5-20 minutes per program. As 3D shapes have a much larger search space, such an approach would not be able to find a solution in reasonable time.
> 2) Stochastic search: Here the problem would be at least as tough as doing inverse graphics, so we can safely assume that this would work no better than MCMC for inverse graphics. In Picture (Kulkarni et al. [2]), their approach takes minutes for a 2D image with simple contours.
>
> We have contacted the authors of these two papers, who confirmed our estimates of the efficiency of their methods.
>
> [1] Ellis, Kevin, Armando Solar-Lezama, and Josh Tenenbaum. "Unsupervised learning by program synthesis." NIPS 2015.
> [2] Kulkarni, Tejas D., et al. "Picture: A probabilistic programming language for scene perception." CVPR 2015.
>
> 3. Decomposition
> Thanks for the positive comment on the decomposition. The results just correspond to top-1 predictions.
>
> 4. Interpreter
> Our semantic operators correspond to simple geometric primitives. Therefore, it’s quite straightforward to write an interpreter for them. The programs in our DSL are tokenized vectors and can be directly feed into the neural program executor. Adding new semantic operator to the DSL is thus easy. We just need to re-train or finetune the current program executor with the new semantic operator included.
>
> We have also listed all other planned changes in our general response above. Please don’t hesitate to let us know for any additional comments on the paper or on the planned changes.

---

> > ### Author Response · Authors · 2018-12-03
> > **Revision Uploaded**
> >
> > Dear Reviewer 2,
> >
> > Thanks again for your constructive comments. We have made substantial changes in the revision according to the reviews. In particular, we have compared our model with three additional baselines, including CSGNet, in Table 2 and Sec 5.2. We’ve also discussed the design of DSL and search-based models (Sec 6).
> >
> > As the discussion period is about to end, please don’t hesitate to let us know if there are any additional clarifications that we can offer. Thanks!

---

### Official Review · AnonReviewer1 · 2018-11-03
**Good paper!**

**Rating:** 7
**Confidence:** 5

**Review:**

This paper introduces a high-level semantic description for 3D shapes. The description is given by the so-called ShapeProgram,  Each shape program consists of several program statements. A program statement can be either Draw, which describes a shape primitive as well as its geometric and semantic attributes, or For, which contains a sub-program and parameters specifying how the sub-program should be repeatedly executed. The ShapeProgram is connected with an input through two networks, the program generator (encoder) and a neural program executor (decoder). Both encoder/decoder are implemented using LSTM. The key ML contribution is on the decoder, which leverages a parametrization to make the decoder differentiable. The major advantage of the proposed technique is that it does not need to specify the ShapeProgram in advance. In the same spriit of training an auto-encoder. It can be learned in a semi-supervised manner. However, in practice, one has to start with a reasonably good initial program. In the paper, this initial program was learned from synthetic data.

The paper presents many experimental results, including evaluation on synthetic datasets, guided adaptation on ShapeNet, analysis of stability, connectivity measurement, and generalization, and application in shape completion. The presented evaluations, from the perspective of proposed experiments, is satisfactory.

On the downside, this paper does not present any baseline evaluation, party due to the fact that the proposed problem is new. In fact, existing inverse procedural modeling techniques require the users to specify the program. However, the proposed approach could be even more convincing if it evaluates the performance of semantic understanding. For example, would it be possible to evaluate the performance on shape segmentation?

Additional comments:
1. How important is the initial program?

2. The interactions among shape parts usually form a graph, not necessarily hierarchical. This should be discussed.

3. What is the difference between 3D shapes and 3D scenes? Does this approach require a front/up-right orientation?

4. It would be interesting to visualize/analyze the intermediate representations of the neural shape generator. Does it encode meaningful distributions among shape parts?

Overall, it is a good paper, and I would like to see it at ICLR 2019.

---

> ### Author Response · Authors · 2018-11-15
> **Response to Reviewer 1**
>
> Thank you very much for the constructive comments.
>
> 1. Baselines
> We agree that it’s important to add more baselines. In the revision, we will include comparisons with the following three algorithms:
> 1) Nearest neighbors. For a given test shape, we search its nearest neighbor in the training set.
> 2) CSGNet-original (the original model released by the authors of CSGNet)
> 3) CSGNet-augmented (the augmented CSGNet model trained on our dataset with additional shape primitives we introduced).
>
> Evaluating on shape segmentation is definitely an interesting direction. We’ve started working on it. As data processing takes additional time, we’ll either include the results into the revision by Nov 23 or, if it’s not done by then, into a later revision.
>
> 2. Specific Questions
> (1) Initial programs
> The initial synthetic programs provide supervised bootstrapping to initialize the program synthesis network. These programs are essential: we observe that without bootstrapping the model cannot converge to a meaningful point. They, however, can be very simple: e.g., 10 simple table templates (Fig. A1) are sufficient to initialize the model, which later achieves good performance under execution-guided adaptation.
>
> (2) Interaction
> Thanks! We agree that the graphs are a more general representation for object parts and can be important next steps. We’ll include this into discussion as suggested.
>
> (3) Shapes vs scenes
> Compared with scenes, 3D shapes more frequently have program-like regularities, such as repetition and symmetry. An interesting future direction is to explore how programs can be used to explain scenes. Our current model requires a front and up-right orientation.
>
> (4) Visualization
> As suggested, we will manipulate the representation after the LSTM to see how different dimensions affect the generated shape primitives.
>
> We have also listed all other planned changes in our general response above. Please don’t hesitate to let us know for any additional comments on the paper or on the planned changes.

---

### Official Review · AnonReviewer3 · 2018-11-04
**Addresses an important problem; well written; but missing baselines and some discussions**

**Rating:** 6
**Confidence:** 4

**Review:**

This paper presents an approach to infer shape programs given 3D models. The programs include placing and arranging predefined primitives in layouts and can be written as a program over a domain-specific language (DSL).

The architecture consists of a recurrent network that encodes a 3D shape represented as a voxel grid and outputs the instructions using a LSTM decoder. The generation is two-step where the first step predicts a program ID and the second step predicts instructions within the program ID. This aspect wasn't completely clear to me, see questions below. A second module that renders the program to 3D is also implemented as a neural network in order to optimize the model parameter in a end-to-end manner by minimizing a reconstruction loss.

The method is evaluated on 3D shape reconstruction tasks for chairs and tables categories of the ShapeNet dataset. The approach compares favorably to Tulsiani et al., which considers a shape to be composed of a fixed number of cuboids.

The paper is well written and investigates an important problem. But it is hard to tease of the contributions and the relative importance of various steps in the paper:

1. Structure search vs. prediction. How does the model perform relative to a search-based approach for program generation. That would be slower but perhaps more accurate. The prediction model can be thought of an amortized inference procedure for search problems. What advantages does the approach offer?

2. Choice of the DSL. Compared to CSG modeling instructions of Sharma et al. the proposed DSL is more targeted to the shape categories. While this restricts the space of programs (e.g., no intersection, subtraction operations are used) leading to better generation of chairs and tables, it also limits the range and generalization of the learned models to new categories. Some discussion and comparison with the choice of DSL would be useful.

3. Is the neural render necessary -- Wouldn't it be easier to simply use automatic differentiation to compute gradients of the rendering engine?

4. It is also not clear to me how having a differentiable renderer allows training in an end-to-end manner since the output space is discrete and variable length. In CSGNet (Sharma et al.) policy-gradient techniques were used to optimize the LSTM parameters. The details of the guided adaptation were unclear to me (Section 4.3).

5. Is the neural renderer reliable -- Is is not clear if the neural renderer can provide accurate gradients when the generated programs are incorrect since the model is trained on a clean samples. In practice this means that the encoder has to initialized well. Since the renderer is also learned, would it generalize to new programs within the same DSL but different distribution over primitives -- e.g., a set of tables that have many more legs. Some visualizations of the generated shapes from execution traces could be added, sampling programs from within and outside the program distributions used to train.

6. All the above points give an impression that the choice of DSL and careful initialization are important to get the model to work. Some discussion on how robust the model is to these choices would be useful. In other words how meaningful is the generalization from the supervised training set of templates chairs and tables?

7. Missing baselines: The model is trained on 100,000 chairs and tables with full supervision. What is the performance of a nearest neighbor prediction algorithm? This is an important baseline that is missing. A comparison with a simplified CSGNet with shape primitives and union operations is also important. Tulsiani et al. consider unions but constrain that all instances have the same number of primitives which can lead to poor reconstruction results. Furthermore the training sets are likely different making evaluations unclear. I suggest training the following decoders on the same training set used in this approach (1) fixed set of cuboids (e.g., Tulsiani et al.), (2) A recurrent decoder with cuboids, (3) CSGNet (different primitives and operations), (4) a nearest neighbor predictor with the Hamming or Chamfer distance metric.

---

> ### Author Response · Authors · 2018-11-15
> **Response to Reviewer 3 (Part 2)**
>
> 4. Data-efficiency, initialization, and robustness
> Our model is data-efficient. It’s trained on 100K chairs and tables, but without supervision. The only supervision it requires is the small number of shape templates, which are used for initializing the program generator. We agree with the reviewer that such initialization is essential: we observe that without bootstrapping the model cannot converge to a meaningful point. They however can be very simple: e.g., 10 simple table templates (Fig. A1) are sufficient to initialize the model, which later achieves good performance under execution-guided adaptation. Our model is also robust: it works well after pre-training on these 10 simple templates, with and without the semantic meaning of DSL. It also generalizes to shapes from unseen categories, as shown in Sec 5.4.
>
> We have also listed all other planned changes in our general response above. Please don’t hesitate to let us know for any additional comments on the paper or on the planned changes.

---

> ### Author Response · Authors · 2018-11-15
> **Response to Reviewer 3 (Part 1)**
>
> Thank you for the thoughtful review.
>
> 1. Baselines and structured search
> Thanks for the suggestion! We agree that it’s important to add more baselines. We clarify that the current result from Tulsiani is already from a re-trained model. In the revision, we will include additional comparisons with the following three algorithms:
> 1) Nearest neighbors. For a given test shape, we search its nearest neighbor in the training set.
> 2) CSGNet-original (the original model released by the authors of CSGNet)
> 3) CSGNet-augmented (the augmented CSGNet model trained on our dataset with additional shape primitives we introduced).
>
> Amortized inference is essential for our task due to its large search space. Our model takes 5 ms to infer a shape program with a Titan X GPU. There are two possible approaches for a structured search over the space of programs, both of which would be too slow for our task:
> 1) Constraint solving: we would have to use an SMT solver. Ellis et al [1] used SMT solvers to infer 2D graphics programs, and takes on the order of 5-20 minutes per program. As 3D shapes have a much larger search space, such an approach would not be able to find a solution in reasonable time.
> 2) Stochastic search: Here the problem would be at least as tough as doing inverse graphics, so we can safely assume that this would work no better than MCMC for inverse graphics. In Picture (Kulkarni et al. [2]), their approach takes minutes for a 2D image with simple contours.
>
> We have contacted the authors of these two papers, who confirmed our estimates of the efficiency of their methods.
>
> [1] Ellis, Kevin, Armando Solar-Lezama, and Josh Tenenbaum. "Unsupervised learning by program synthesis." NIPS 2015.
> [2] Kulkarni, Tejas D., et al. "Picture: A probabilistic programming language for scene perception." CVPR 2015.
>
> 2. DSL
> We agree that a DSL with semantics has advantages and disadvantages: on one hand, it offers semantic correspondence and enables better in-class reconstructions; on the other hand, it may limits the ability to generalize to shapes outside training classes. Our current instantialization focuses on the semantics of furnitures (which can be viewed as a superclass, whose subclasses share similar semantics). Within this superclass, our model generalizes well: trained on chairs and tables, it generalize to new categories such as “bed”, ‘“bench”, “sofa” and “cabinet” (Sect. 5.4). We’ll include a discussion on the choice of DSL in the revision.
>
> 3. Neural program executor
> Thanks for the comments on the neural program executor. We’ll include the following discussion into the revision to improve the clarity of the paper.
>
> A) Automatic differentiation
> Our program executor takes as input a tokenized program and produces a voxelized 3D primitive. Due to the use of high-level program sentences such as `for’, there is no explicit differentiable formula for such process. We therefore use a neural network to approximate it.
>
> B) End-to-end training
> The output of the program inference mode is continuous (continuous probability over tokenized programs and continuous parameters). After getting the output of the program inference model, a real execution engine (not the neural executor) contains two steps (1) discretization such output  and (2) execute the discretized program to generate the voxel. Our neural executor is leaned to jointly approximate both steps, thus the whole pipeline can be differentiable end-to-end. We apply max-pooling over all of the blocks; therefore, the system can handle a variable number of blocks and still be differentiable.
>
> C) Reliability
> We agree that a typical concern regarding a neural executor is on their generalizability to input outside the training distribution. This is also the underlying motivation behind our design---we train a program executor that operates on block-level programs, not full shape programs. While it’s hard to cover all possible shape programs in training, covering the distribution possible block-level programs is easy (e.g., tables with many legs), as they have a smaller degree of freedom. In training the executor, we are no longer concerned about the possible combination of different blocks. Such a decomposition allows the executor to guide the program synthesizer/generator to generalize to new programs that are not in the training distribution: while the synthetic tables only contains 10 different combinations of block programs, the guided adaptation with the extensively learned neural executor allows our model generalize to other unseen combinations of block programs. In fact, Fig 5 (c),(d) are newly learned templates beyond the pre-trained templates shown in Fig A1.

---

> ### Comment · Area_Chair1 · 2018-11-30
> **Please respond to author comments and discussion**
>
> Reviewer 3,
>
> The authors have made substantial changes to their paper in response to your and others' reviews, as well as responded to your comments. Please take some time, in the last week of the discussion period, to consider their response, engage in discussion if needed, and either explain why you stand by your assessment or reconsider your score.

---

### Public Comment · (anonymous) · 2018-10-10
**Lack of justification/motivation, no comparison with related works**

The problem studied (3d shape programs) is interesting. However, I think the work presented in this paper may be incremental and lack of theoretical justification and comparison with related works.

There are many existing works out there for solving neural program inference/execution problems (e.g. for 2d shapes, hand-drawings, function execution and inference). They have achieved good performance in their respective experiments, e.g. neural program interpreter (Reed et al). Why is your proposed method better than existing models? There is little or no justification or insight in the paper. The use of LSTM and 3D convolution is fairly standard, so it is fair to say no new paradigm or framework is provided. With the standard architectures/components, as a reader, I would really like to see

1) why is your proposed method better than other related models? Can you provide theoretical justifications and insights? I understand different papers may target slightly different applications, however, the general concept and framework should be transferable to slightly different tasks. (e.g. 2d shape programs should be easily transferable to 3d) Therefore, as a reader, I think theoretical motivations are very important here. Otherwise, it is not clear how difficult the problem is, and what models are really effective for solving it.

2) Lack of experimental comparison. Only performance of the proposed method is shown. Again, can you empirically show that other related models would not perform well in this setting?


3) The shapes in the experiments are somewhat too simple. Only shapes like tables and chairs are considered, and these shapes tend to have fairly simple and clean structures. Can you show that the proposed method can generalize to more complex and diverse shapes?

---

> ### Author Response · Authors · 2018-10-14
> **Our response**
>
> Thank you for the feedback. Please see our response above.

---

### Author Response · Authors · 2018-10-14
**Our response to the earlier reader's comment and some general thoughts**

[Note: This is a reply to the reader's comment below.] We thank the anonymous reader for the feedback, which actually revealed the gap between the views of researchers from different communities. Here we take the chance to reply to these comments in specific, but also present our observation of the gap in general.

Most importantly, the paper is about introducing a new 3D shape representation---shape programs---not about a new model for program synthesis or execution. Modeling 3D shapes is a classic and central problem in computer graphics and computational geometry, where the community have been working on it for decades, introducing various representations such as point clouds, voxels, splines, meshes, and primitives. However, as we emphasized in the abstract and intro, these representations do not capture high-level shape regularities such as repetition and symmetry explicitly, while human perception rely heavily on these cues.

The key contribution of our paper is therefore on proposing shape programs as a new 3D shape representation, along with a practical framework for learning them. The main challenge of introducing a new shape representation is the lack of annotated data. On 2D hand drawings, Ellis et al. solved the problem by having neural nets discover low-level traces for an off-the-shelf program synthesizer, but their approach failed to discover 3D shape programs due to the much larger search space. We instead propose to learn a simple, fast, approximate neural program executor and use it to guide the training of the neural program synthesizer. Having the neural executor in the loop allows fast adaptation to shapes outside training distribution. This includes general shapes without program annotations, as well as shapes from a different category.

We showed that the new shape program representation and the learning paradigm work together to reconstruct shapes well, and capture important shape properties such as stability better than those using representations like voxels or primitives. The specific network architectures used for inference and execution are components of the framework, and can be extended or replaced with more advanced ones without affecting the main message of the paper.

3D shapes are complex; modeling 3D shapes is challenging. Developing an approach that works with the range of 3D object shapes we address here is nontrivial.  The reader’s suggestion that 2D methods ‘can be easily transferable to 3D’ is unjustified and does not fit with the reality in computer vision and computer graphics, where many researchers have spent their careers working on these problems. In particular, the furniture object classes we study are among the largest categories in the main public 3D shape repository, ShapeNet, and have been very widely studied in the computer vision and graphics community due to their complexity (Parsing IKEA Objects, ICCV’13; Joint Embeddings of Shapes and Images, ACM TOG’15; and many others). By computer vision community standards, we consider a range of complex chairs and tables (e.g. Fig A1(b)), and we have also included results on generalizing to new shape categories such as beds, benches, cabinets, and sofas.

Regarding comparison with alternative methods, we have focused on comparisons with state-of-the-art 3D shape reconstruction methods, because our goal has been to show the value of learning and inferring shape programs for 3D shape perception and understanding.  We thus compared with state-of-the-art methods of Tulsiani et al (CVPR’17) and Wu et al (NIPS’17), and we have evaluated our model on the latest most challenging benchmark of real world images and shapes (Sun et al, CVPR’18). Building models that work well on real, in-the-wild images is challenging, and its significance should not be undervalued. We will also include a comparison to CSGNet (Sharma et al, CVPR’18) in the revision.

We also recognize the point that it would be valuable to compare with other general-purpose neural program learning approaches, although as discussed above, it is unlikely that any general approach could be applied simply out of the box, without some adaptation to the specifics of 3D shapes. In our revision, we will highlight ways that our particular approach to representing and learning shape programs is well suited to the challenges of 3D shape modeling, relative to previous methods. In particular, in addition to the idea of ‘execution-guided learning’, we want a recognition model that exploits the fact that (a) objects are made of parts, and (b) parts have program-like regularities in their geometry and their relative arrangement. Before the submission deadline, we had contacted the authors of neural program interpreters a few times for their implementation, but did not receive a reply. If there is any specific algorithm that reviewers think we should compare with, especially if code is available, please let us know and we will try to include a comparison in the revision.

---

> ### Comment · Area_Chair1 · 2018-11-08
> **Notifications**
>
> Just FYI, it is better to separately (or in addition to your current response) reply to the reviewer's reviews, so they get a notification that there is activity on their thread. You may want to leave a quick comment for each reviewer that you've produced a combined response.

---

> > ### Author Response · Authors · 2018-11-08
> > **Thanks for the suggestion**
> >
> > Thank you, AC. We agree and will follow your suggestion. The response here was posted a while ago, and is actually not to the official reviews, but to the earlier public comment. We're still working on the response to official reviews and will post them separately once they are ready.

---

> > > ### Comment · Area_Chair1 · 2018-11-11
> > > **I see**
> > >
> > > I understand that this was in response to an earlier public comment. It would have been better to respond directly to that comment to avoid confusion. While the timestamp should make clear that this is not a response to reviewers, there is still scope for confusion. I recommend editing this comment to add, at the top, that it's a reply to the comment below.

---

> > > > ### Author Response · Authors · 2018-11-11
> > > > **Thanks again.**
> > > >
> > > > Thanks again, AC. We've updated the title of the comment and added a note at the top.

---

### Author Response · Authors · 2018-11-15
**Our General Response**

We thank all reviewers for their comments. In addition to the specific response below, here we summarize the changes planned to be included in the revision.

As suggested by reviewers, we plan to include the following changes in the revision by Nov. 26 (the new official revision deadline, extended from Nov. 23):
- We will cite and discuss the suggested related work.
- We will discuss more about the design of DSL, structure search v.s. amortized inference, etc.
- We will add more baselines, including:
   1) Nearest neighbors. For a given test shape, we search its nearest neighbor in the training set.
   2) CSGNet-original (the original model released by the authors of CSGNet)
   3) CSGNet-augmented (the augmented CSGNet model trained on our dataset with additional shape primitives we introduced).
- We will visualize the intermediate representation of neural shape generator (neural program executor).

Please don’t hesitate to let us know for any additional comments on the paper or on the planned changes.

---

### Author Response · Authors · 2018-11-27
**Summary of Revision**

Dear Reviewers and AC,

Thank you for your constructive comments. We have revised our paper accordingly. The main changes include:

1) We have added more baselines, including the original CSGNet, the augmented CSGNet, and Nearest Neighbours (Section 5.2 and Table 2).
2) We have analyzed the intermediate representation of the shape generator (Section 6 and Figure 8).
3) We have included discussion on the design of the DSL, structure search v.s. amortized inference, and future work (Section 6).
4) We have revised the paper to better explain the end-to-end differentiability of our model (Section 4.2 and A.2) and the role of the initial programs (Section 5.2).

Please don’t hesitate to let us know for any additional feedback. Thanks!

---

### Meta-Review · Area_Chair1 · 2018-12-13
**Good paper**

**Confidence:** 5
**Recommendation:** Accept (Poster)

**Metareview:**

This paper presents a method whereby a model learns to describe 3D shapes as programs which generate said shapes. Beyond introducing some new techniques in neural program synthesis through the use of loops, this method also produces disentangled representations of the shapes by deconstructing them into the program that produced them, thereby introducing an interesting and useful level of abstraction that could be exploited by models, agents, and other learning algorithms.

Despite some slightly aggressive anonymous comments by a third party, the reviewers agree that this paper is solid and publishable, and I have no qualms in recommending it from inclusion in the proceedings.